# A Patient-Ready Wearable Transcutaneous CO_2_ Sensor

**DOI:** 10.3390/bios12050333

**Published:** 2022-05-13

**Authors:** Juan Pedro Cascales, Xiaolei Li, Emmanuel Roussakis, Conor L. Evans

**Affiliations:** Wellman Center for Photomedicine, Massachusetts Genetal Hospital, Harvard Medical School, Charlestown, MA 02129, USA; xli77@mgh.harvard.edu (X.L.); erousakis@mgh.harvard.edu (E.R.)

**Keywords:** CO_2_ partial pressure, transcutaneous sensor, luminescent probe, HPTS, wearable device

## Abstract

Continuously monitoring transcutaneous CO_2_ partial pressure is of crucial importance in the diagnosis and treatment of respiratory and cardiac diseases. Despite significant progress in the development of CO_2_ sensors, their implementation as portable or wearable devices for real-time monitoring remains under-explored. Here, we report on the creation of a wearable prototype device for transcutaneous CO_2_ monitoring based on quantifying the fluorescence of a highly breathable CO_2_-sensing film. The developed materials are based on a fluorescent pH indicator (8-hydroxy-1,3,6-pyrenetrisulfonic acid trisodium salt or HPTS) embedded into hydrophobic polymer matrices. The film’s fluorescence is highly sensitive to changes in CO_2_ partial pressure in the physiological range, as well as photostable and insensitive to humidity. The device and medical-grade films are based on our prior work on transcutaneous oxygen-sensing technology, which has been extensively validated clinically.

## 1. Introduction

Carbon dioxide (CO_2_) plays various roles in the human body such as the respiratory drive, regulation of blood pH, and affinity of hemoglobin for oxygen [1]. Monitoring CO_2_ partial pressure ($pCO_{2}$) in breath and blood is of great importance for both the medical diagnosis and treatment of human diseases such as respiratory and metabolic disorders. For example, the adequacy of spontaneous and mechanical ventilation is usually evaluated by measuring the CO_2_ concentration in arterial blood [2,3]. However, the current “gold standard” method to obtain this reading relies on the invasive process of arterial blood gas sampling through the placement of arterial lines [2,3]. Although there exist other non-invasive, albeit indirect, clinical alternatives for CO_2_ sensing, these techniques require further development and optimization to improve accuracy, integration into clinical workflows, etc., as they exhibit important limitations or drawbacks. For example, end-tidal CO_2_ sensors use an infrared absorption technique for measuring the level of CO_2_ released at the end of an exhaled breath [4]. However, this technique requires the use of an optical cavity, requiring large working volumes of gas compared to other methods [5], and hence, its accuracy to estimate arterial $pCO_{2}$ is susceptible to sampling errors and patient-related factors (e.g., age, patient positioning, lung disease) [6,7].

A non-invasive alternative to arterial gas sampling is to monitor $pCO_{2}$ on the skin surface, which can reduce or altogether eliminate the need for blood gas sampling, decreasing the risk for patient co-morbidities and improving patient comfort. Transcutaneous monitoring of CO_2_ can be critical, for example, to assess ventilation in neonates [8], for which periodic arterial blood sampling can be painful and does not provide continuous readings, and monitoring end-tidal CO_2_ is not possible due to the small tidal volumes. There are many different locations on the body that can be used for transcutaneous monitoring, depending on the clinical scenario, on which transcutaneous $pCO_{2}$ is highly correlated to arterial $pCO_{2}$, typically highly vascularized areas with thin skin. Some examples include the earlobe [9], the neck near the carotid artery [10], the lateral abdomen, the anterior or lateral chest [11], the volar forearm, the inner upper arm or the inner thigh [12], the foot [13], etc. However, current technology based on electrochemical sensors requires large and expensive equipment, long bed-side calibration procedures, immobile patients, etc. [3,14]. Therefore, it would be of considerable interest to develop miniaturized sensors that could be incorporated into wearable devices and find wide application, such as the continuous measurement of transcutaneous CO_2_.

Optical transcutaneous CO_2_ sensors based on luminescent materials may offer several advantages, such as accurate detection of CO_2_ levels, as well as great potential for miniaturization [15]. Such sensors have traditionally employed a pH indicator that exhibits different fluorescent intensities upon exposure to different CO_2_ concentrations [16,17,18]. The pH-sensitive fluorescent dye 8-hydroxy-1,3,6-pyrenetrisulfonic acid trisodium salt (HPTS) is one of the most widely used in optical CO_2_ sensors [19,20]. For skin-worn devices, it can be important to create sensors whose response will not be altered by changes in humidity [21,22], which can vary widely depending on climate or body location [23]. In order to make HPTS molecules compatible with hydrophobic matrices, a lipophilic hydrated ion pair is usually formed by converting the dye into its anion form with a quaternary ammonium cation [16,17,19]. In addition, a phase transfer reagent (quaternary ammonium hydroxide) co-embedded along with the dye within a support matrix is necessary to facilitate the tuning of the materials’ sensitivity and enhanced stability. Therefore, the performance of the CO_2_ sensors depends not only on the properties of the dye molecule, but also on the optical and physical properties of the support matrix.

In this manuscript, we report on the development of polymer films with embedded HPTS-based ion pairs, providing highly breathable materials that exhibit bright emission throughout the physiological CO_2_ range. We created a wireless and non-invasive wearable prototype that, in conjunction with the proposed materials, aims to continuously monitor transcutaneous CO_2_ partial pressure.

## 2. Materials and Methods

### 2.1. Materials

HPTS, tetraoctylammonium bromide (TOABr), tetraoctylammonium hydroxide solution (20% in methanol) (TOAOH), hexadecyltrimethylammonium hydroxide solution (25% in methanol) (CTAOH), poly(methyl methacrylate) (PMMA) (Approx Mw 75,000), platinum (0)-1,3-divinyl-1,1,3,3-tetramethyldisiloxane complex solution, and sodium sulfate were purchased from Sigma-Aldrich. Cetyltrimethylammonium bromide (CTABr) was purchased from Fisher Scientific. Poly(propyl methacrylate) (PPMA) was purchased from Scientific Polymer Products (Appox Mw 150,000). The white pigment concentrate, (45–55% methylhydrosiloxane)-dimethylsiloxane copolymer (HMS), and cure-retarding agent were purchased from Gelest. Glass microfiber filters were purchased from Whatman and the medical-grade adhesive films (Bioclusive) from McKesson.

### 2.2. Synthesis of Ion Pairs

#### 2.2.1. Synthesis of (HPTS)/(CTA)_3_

This compound was synthesized by ion-pairing HPTS with CTABr, which was similar to the approach adopted by Burke et al. [24]. Eighty milligrams of CTABr was dissolved in 5 mL of ultrapure water at 50 °C and mixed with a solution comprising 40 mg of HPTS in 5 mL of ultrapure water. The product was obtained by vacuum filtration followed by washing with ultrapure water. The solid product was dried in an oven at 50 °C for 1 h.

#### 2.2.2. Synthesis of (HPTS)/(TOA)_4_

The (HPTS)/(TOA)_4_ was prepared by the following method [25]: 20 mg of HPTS and 4-fold molar equivalents of TOABr (85 mg) were dissolved in 5 mL of 0.01 M NaOH solution and 5 mL of dichloromethane, respectively. The 2 solutions were subsequently mixed together, and the reaction mixture was stirred for about 1 h at room temperature. The mixture was added into a separatory funnel, and the ion pair was extracted into the organic layer followed by washing twice with 5 mL 0.01 M NaOH solution. The organic layer was collected and dried from traces of water over sodium sulfate. The solvent was removed by rotary evaporation, and the solid product was dried under high vacuum. The product yield was calculated to be about 60%.

### 2.3. CO_2_-Sensing Film Preparation

#### 2.3.1. Films for Spectral Characterization

Filter paper was used as the substrate material to produce samples for spectral characterization, as it is highly breathable and provides light scattering, enhancing the amount of collected light by our spectrometer. An aliquot of ion pair solution was added into 0.05 mg/μL PMMA or PPMA in dichloromethane followed by different ratios of methanolic CTAOH or TOAOH and mixed thoroughly by vortexing. Then, 60 μM (HPTS)/(CTA)_3_ and 240 or 480 μM (HPTS)/(TOA)_4_ were applied to the final solutions. Approximately 15 μL of that solution was deposited onto the 5 mm-diameter filter paper, which was placed on a solid surface and was left to dry in the hood overnight at room temperature.

#### 2.3.2. Multilayer CO_2_ Sensing Film for the Wearable

As shown in Figure 1, the multi-layer CO_2_-sensing film was fabricated following the approach in [26], by stacking a breathable and white (scattering) silicone film, a PPMA-based sensing film, and a transparent semipermeable film (Bioclusive, McKesson, New York, NY, USA). This configuration was used along with the wearable due to its reduced volume and thickness and, therefore, fast equilibration to the skin CO_2_ concentration. The combination of the white silicone layer/PPMA layer was used in place of filter paper in order to minimize the dead-space volume while allowing for the fluorescence emission to be backscattered to the wearable’s photodiode. The white coating also serves as an optical insulation, preventing external lighting from affecting the measurement, yielding a reading that is independent of skin tone. The semi-permeable, optically transparent adhesive film was used to seal out room air from the CO_2_-sensing film and white silicone layer, allowing the material to equilibrate to skin $pCO_{2}$. Our approach to transcutaneous CO_2_ sensing is biocompatible since only the multilayer film makes contact with the skin. Of the total area of the film, close to 99% corresponds to a commercial, medical-grade, skin adhesive and the remaining 1% to the white silicone coating, which is compatible with skin [26] and can even be inserted into tissue [27]. The HPTS/PPMA disk is prevented from direct contact with the skin via the white coating.

To prepare the PPMA-based film, a 25 μL solution of PPMA with (HPTS)/(TOA)_4_ and TOAOH was deposited into an 8mm-diameter circular mold on a glass slide. The PPMA-based film was removed from the glass slide after drying in the hood for 30 min. The white scattering layer was prepared following the protocol in [26] with the following modifications. The silicone polymer component (100 μL) was first mixed with the white pigment concentrate (1 g), and then, 3 drops of cure retarding agent and 1 drop of platinum catalyst were added. The mixture was deposited on a flat surface and was able to form a thin film before curing.

### 2.4. Principle of Operation

The most common principle for optical CO_2_ sensors is based on fluorescence changes of pH indicators upon their protonation or deprotonation at different pH values [17,18]. In HPTS, it has been widely reported that the pH sensitivity of HPTS arises from the OH group [19]. To facilitate the dissolution of the dye in the hydrophobic polymers, the lipophilic quaternary ammonium cation (Q^+^) (known as the phase transfer agent) is widely used to form ion pairs with the pH indicator anion (D^−^) and provides the water required for the production of carbonic acid to protonate the pH indicator dye [16]. The general chemical principle can be described as follows: (1)$$Q^{+}D^{-}.xH_{2}O+CO_{2}⇌Q^{+}HCO_{3}^{-}.\left(x-1\right)H_{2}O+DH$$

A number of water molecules are found to be associated with the ion pair when a phase transfer agent is used to extract an anion indicator from an aqueous solution into an organic solution [18]. In the presence of CO_2_, the dye anion with green fluorescence is converted into its protonated form, which does not fluoresce under 470 nm excitation. The reaction of the indicator dye with CO_2_ is fully reversible. In addition, the quaternary ammonium hydroxide is introduced in order to tune the sensitivity of the materials.

### 2.5. Fluorescence Spectral Measurements

Fluorescence spectral measurements were acquired using an FLS1000 Steady State and Luminescence Lifetime Spectrometer equipped with a continuous xenon lamp (Xe2) (Edinburgh Instruments, Livingston, UK). The sample tested was placed on a holder affixed diagonally inside a cuvette with a septum screw cap. Changes in gas conditions were generated by flowing a N_2_/CO_2_ gas mixture via a needle through the septum of the cuvette cap. The gases were also connected to a water bubbler to modify the water vapor content of the mix. Excitation spectra were acquired by setting the emission wavelength at 570 nm and scanning the excitation wavelength from 300 nm to 550 nm. Emission spectra were acquired by setting the excitation wavelength at 405 or 470 nm and scanning the emission wavelength from 500 nm to 700 nm. For the emission spectra, the excitation light was filtered using a 495 nm long-pass filter for all the samples, which is consistent with the wearable’s filters and therefore provides a “ground truth” measurement. The photostability of the different materials was measured using a Kinetic Scan with a fixed excitation wavelength at 470 nm and a fixed emission wavelength at 520 nm for a time period of 120 min under air conditions. The power of the excitation light that the samples were exposed to was set to approximately 0.01 mW by adjusting the excitation bandwidth and was measured by an optical power meter (Thorlabs, PM100D, Newton, NJ, USA). The percent change of the intensity was calculated as follows:(2)$$I\%=\frac{I-I_{min}}{I_{max}}\cdot 100$$

### 2.6. Wearable Optical Device

Based on our previous efforts [26], we developed a small and lightweight wearable prototype (see Figure 1a), which can provide readings of the partial pressure of $CO_{2}$ by exciting and detecting the fluorescent response of the $CO_{2}$-sensing films described above.

The devices are based on a WiFi-enabled microcontroller board (Particle Photon), which drives the low-power, custom electronics (a fast ADC chip, transimpedance amplifier, and signal-conditioning block). The intensity of the fluorescence response of the film is excited sequentially by two high-power LEDs with peak wavelengths of 405 and 470 nm (see Figure 1b) and detected via a PIN photodiode. The LEDs are modulated by a sinusoidal voltage signal of f=1.6 kHz, and the intensity of the emission from the CO_2_-sensing film excited by each of the LEDs is defined as the amplitude of the measured sinusoidal response, which is extracted via multiple linear regression [26]. The temperature of the film and sensor head was sampled through a small thermistor. To avoid cross-talk between excitation and emission and to remove an unwanted phosphorescence from the LED’s (see Appendix A), the LED excitation was filtered by a 500 nm short-pass filter composed of two ultra-thin flexible optical notch filters (Edmund Optics), which is in turn were blocked by a 500 nm long-pass filter covering the PIN photodiode, by combining a flexible 405 nm long-pass filter (Edmund Optics) and an “amber” color polyamide film (Kapton tape, 3M). The CO_2_ sensing films were attached onto the device’s 3D-printed casing using thin, highly adhesive double-sided tape. The device adheres to the skin via the medical-grade film and an elastic band or strap, which is minimally tightened to prevent the restriction of blood flow. Data were collected via a Python [28] script on a PC through a USB serial port in order to record the sinusoidal time series each time CO_2_ is sampled. For calibrated devices, the calibration algorithm (described below) can be loaded onto the firmware, so the devices directly report CO_2_ readings via USB or WiFi.

## 3. Results and Discussion

### 3.1. Optimization of Sensing Film Compositions

To test the effect of different dyes and polymer matrices on the sensitivity of the CO_2_ sensors, the following materials cast on filter paper were investigated: (1) (HPTS)/(CTA)_3_ in PPMA; (2) (HPTS)/(TOA)_4_ in PPMA; and (3) (HPTS)/(TOA)_4_ in PMMA. The filter paper was used as a support material for spectral measurements, as it is highly scattering and does not change the sensing properties of the materials. Fluorescence spectral changes upon excitation at 470 nm are shown in Figure 2b. The ratio of I_N_2__/I_CO_2__ was used as an indicator of the sensitivity of the CO_2_ sensors, where I_N_2__ and I_CO_2__ correspond to the peak intensities of the spectra when the materials are exposed to a 100% N_2_ and a 100% CO_2_ environment, respectively. It was found that the I_N_2__ of (HPTS)/(CTA)_3_ embedded in PPMA was about 1.7-fold stronger than the I_CO_2__. The CO_2_ sensitivity dramatically increased to about 46 when the sensing molecule was replaced by the (HPTS)/(TOA)_4_. In addition, it was observed that when (HPTS)/(TOA)_4_ embedded in PMMA, I_N_2__ was about 19-fold stronger than I_CO_2__. These results demonstrated that the sensitivity of the CO_2_ sensors could be tuned by adjusting the composition of the sensing molecules and the support matrices.

As has been previously shown for sensors embedded within similar types of synthetic acrylate polymer matrices [22], excitation spectra can provide useful information on the distribution of the embedded sensor molecules and the homogeneity of the resulting materials. Chemical compatibility between the embedded dyes and polymer matrix components featuring hydrophobic functional groups and alkyl chains can have a major impact on sensing performance [21,22]. To gain more insight into the differences in sensitivity observed among the materials tested here that showed (HPTS)/(TOA)_4_ + PPMA exhibiting the highest sensitivity, excitation spectra were acquired and presented in Figure 2c. For all three materials, the dye exhibited maximum excitation at about 400 nm and 500 nm under CO_2_ and N_2_ conditions, respectively. The different excitation wavelengths represent the protonated or deprotonated forms of the dye in the presence of CO_2_ or N_2_. It was found that (HPTS)/(CTA)_3_ in PPMA along with CTAOH was only partially converted to the protonated form under the CO_2_ condition, which explains the low CO_2_ sensitivity of this material. In contrast, (HPTS)/(TOA)_4_ embedded in PPMA along with TOAOH was almost completely converted to the protonated form under the CO_2_ condition, which could result in the observed much higher sensitivity. In addition, it can be clearly seen that the excitation spectrum of (HPTS)/(CTA)_3_ in PPMA along with CTAOH is much broader than that of (HPTS)/(TOA)_4_ in PPMA along with TOAOH under both the CO_2_ and N_2_ conditions. It is worth noting that, while CTAOH bears a single C16 alkyl chain, TOAOH has four octyl (C8) chains, which make it overall more lipophilic. Taken together, these observations suggest there is a poor homogeneity of (HPTS)/(CTA)_3_ in PPMA along with CTAOH, likely leading to effects such as sensor ion pair aggregation, while the most optimally performing material is a combination of the more lipophilic quaternary ammonium ion and phase transfer reagent (TOA^+^/TOAOH) along with the polymer matrix with the longer and more hydrophobic side chain (PPMA). Moreover, it is likely that the relatively longer alkyl side chains of PPMA increase the diffusion and solubility of CO_2_ gas in the matrix, which allows the dye to be protonated more efficiently and ultimately increases the CO_2_ sensitivity, as has been previously suggested [29].

Since (HPTS)/(CTA)_3_ in PPMA along with CTAOH showed much lower sensitivity, subsequent characterizations were only focused on the other two materials. Due to the water-mediated sensing chemistry, the moisture levels of the gas potentially have a strong influence on the sensor performance, particularly in applications such as respiratory gas analysis [16]. Therefore, the CO_2_ sensor performance was further assessed under humid conditions (Figure 2d). It was found that, while the presence of humidity did not affect the sensitivity of (HPTS)/(TOA)_4_ in PPMA + 5% TOAOH towards CO_2_ sensing, the corresponding sensitivity for (HPTS)/(TOA)_4_ in PMMA + 5% TOAOH increased by 1.9-fold. It is possible that the longer alkyl side chains of PPMA make it more hydrophobic and better resistant to water when compared to PMMA. Lastly, although the PMMA-based material showed better photostability during two hours of continuous illumination (Figure 2e), higher gas sensitivity and better water resistance make (HPTS)/(TOA)_4_ in PPMA an optimal candidate for final characterization.

### 3.2. The Effect of TOAOH Ratios on Sensitivity, Photostability, and Dark Stability

After identifying the optimal combination of polymer matrix and sensing dye, the effect of different amounts of TOAOH on the CO_2_ sensing performance was investigated (Figure 3). TOAOH is critical for CO_2_ sensing, as films prepared from the solution without this base buffer yielded no sensitivity to CO_2_. It was also found that (HPTS)/(TOA)_4_ in PPMA exhibited the highest sensitivity with the addition of a 5% (*v*/*v*) methanolic solution of TOAOH. In addition, the CO_2_ sensitivity decreased with the further increase of the TOAOH ratios. A similar trend was observed when a different dye concentration was used (Figure 3a).

To characterize the photostability of the materials, the percentage changes in intensity were obtained after continuous irradiation for 120 min in air. This illumination time represents potentially days or weeks of measurement time with the wearable, depending on the sampling rate, as the acquisition of the CO_2_ concentration lasts only 200 milliseconds per measurement. As shown in Figure 3b, (HPTS)/(TOA)_4_ (240 μM) in PPMA showed higher photostability at higher TOAOH ratios. The percentage intensity changes were found to be 42%, 30%, and 24% for the sensing films made of 240 μM (HPTS)/(TOA)_4_ in PPMA with the addition of 5%, 10%, and 20% (*v*/*v*) TOAOH solution. In addition, 480 μM dye was observed to be less photostable than 240 μM dye in the same support matrix. Although higher TOAOH ratios can improve the materials’ photostability, it decreased the CO_2_ sensitivity and also caused (HPTS)/(TOA)_4_ in PPMA to become less water resistant. It was found that the CO_2_ sensitivity increased by 1.4-fold under humid conditions with the addition of 10% (*v*/*v*) TOAOH (Figure 3c), whereas only 1.1-fold increased sensitivity was observed for 5% (*v*/*v*) addition of TOAOH (Figure 2d). The storage stability of (HPTS)/(TOA)_4_ in PPMA with the addition of 5% and 10% TOAOH was tested over three months. No significant decrease of sensitivity was observed when the materials were stored under ambient and dark conditions for seven days. However, both materials lost about 90% sensitivity after three months (Appendix A). The stability of the materials could be further improved by optimizing the storage conditions.

Therefore, considering its high CO_2_ sensitivity and optimized water resistance and photostability, the formulation (HPTS)/(TOA)_4_ in PPMA with 5% TOAOH was tested in combination with a wearable optical device.

### 3.3. Response and Calibration of the Wearable

As mentioned above, the wearable collects the emission after the dye molecules are excited at two different wavelengths, 405 nm and 470 nm, as was done in [25]. The excitation spectra of the HPTS/TOA-PPMA formulation revealed an isosbestic point around 405 nm in the CO_2_ range of interest (0–50 mmHg), as shown in Figure 4a. This wavelength is therefore an ideal reference or normalization factor to account for variations in film brightness, such as photobleaching, changes in relative positioning between film/device due to motion, etc. The second excitation wavelength used was 470 nm, which yielded a CO_2_-dependent emission from the films. This wavelength was also chosen to allow for sufficient spectral separation between the excitation light (470 nm) and the dye’s emission (520–530 nm) in order to reduce LED leakage into the photodiode. The use of two excitation wavelengths, with the excitation at 405 nm effectively acting as an isosbestic point in our CO_2_ range of interest, allowed us to employ a ratiometric approach, which yielded a metric that was proportional to the CO_2_ concentration and was normalized, not subject to photobleaching effects, and robust against motion artifacts within some range.

Therefore, we define the fluorescence ratio as:(3)$$R=I_{405}/I_{470}$$

As a gold-standard reference, we obtained the fluorescence spectra of TOA/HPTS in PPMA within the multilayer film structure, including a white scattering layer. The measurements were taken by exciting at 405 and 470 nm, with the emission passed through a 495 nm long-pass filter to remove the excitation light from the collected light. To better compare spectrometer measurements with the wearable readings, $I_{405}$ and $I_{470}$ were calculated by integrating the spectra excited at 405 and 470 nm, rather than defining these as the fluorescence intensity at a given wavelength. The integrated signal was smoother and more representative of the signal measured by the wearable’s photodiode, which contains contributions from a wide range of wavelengths (see Appendix A).

Figure 4b plots the fluorescence ratio *R* as a function of the CO_2_ concentration (calculated from the known mix of CO_2_ and N_2_ gases) in a PPMA/white coating sample in which CO_2_ either diffused freely through PPMA (CO_2_ → PPMA) or was forced to do so through the white coating (CO_2_ → White). We allowed one minute for each gas mix to flow before the spectrum was taken. The ratio *R* was normalized between [0, 1] to better compare the trend at different temperatures. When CO_2_ directly diffuses into the PPMA layer (CO_2_ → PPMA), the sample exhibits a cyclical response (start and end points match) that is independent of temperature. The small delay in response speed between increasing and decreasing CO_2_ has been previously reported [25,30] as being due to a difference in the reaction speed between protonation and de-protonation. The diffusion of CO_2_ to the PPMA layer through the white coating (CO_2_ → White) was achieved by adding a transparent, but non-breathable backing layer to force all diffusion to occur through the white coating. As is shown in Figure 4b, the white coating adds a considerable delay in the diffusion of CO_2_ out of the film at 25 °C, not found to happen with oxygen [26]. The breathability of the white film was greatly enhanced by increasing the temperature above 40 °C, as is also seen in the wearable measurements below.

Measurements with the wearable were carried out in a sealed chamber, where a controlled mix of N_2_ and CO_2_ gases was fed in using a programmable gas proportioner (Gometrics) as shown in Appendix A. The $pCO_{2}$ in the chamber was measured with a commercial non-dispersive infrared (NDIR) CO_2_ sensor (K-33 BLG, CO_2_ Meter, Ormond Beach, FL, USA). We found a small amount of LED emission leaks through the optical filters and in the photodiode signal. We quantified this leakage as a function of temperature by measuring with the wearable a “blank” film, i.e., a multilayer film containing all the described layers, but no HPTS dye. This LED leakage was subtracted from all the $I_{405},I_{470}$ values used to calculate *R*. Figure 4c,d shows the fluorescence ratio change due to the same gas mix sequence, measured at an average temperature of 25 °C and 44 °C. The lower temperature was found to lag with decreasing CO_2_, while the higher temperature showed no delay. Further, the normalized R vs. $pCO_{2}$ loops shown in Figure 4e displayed the same trend with temperature as shown in the spectral measurements from Figure 4b. Figure 4f plots the temperature dependence of the delay or lag between the wearable and the reference commercial sensor, obtained from the cross-correlation of both signals, which clearly shows how the white scattering layer becomes significantly more permeable to CO_2_ above 40 °C. The dashed line corresponds to the fit of a second-order polynomial and is shown as a guide to illustrate the trend.

In order to calibrate our prototype, as well as test for cyclability, we programmed a multiple, 5 min-long, cycle sequence into the gas mixer. The measurement shown in Figure 5a was carried out at an average temperature of 45 °C. The fluorescence ratio showed a sensitive and fast response to changes in CO_2_ during the 90 min measurement.

Following the work in [25], the dependence of $pCO_{2}$ on the fluorescence ratio *R* obeys the following equation:(4)$$pCO_{2}=\frac{A\cdot R-B}{-C\cdot R+D}$$
where the constants (A,B,C,D)>0 are obtained by combining different kinetic rates and material parameters such as quantum yield, etc. (see [25]). We fit Equation (Equation 4) to the data in the figure using a script written for GNU Octave [31]. As can be seen in Figure 5b, the model captures the trend well ($R^{2}=0.9518$) and produces a $pCO_{2}$ estimate (Fit1 in Figure 5c), which quantitatively follows the reference sensor’s readings. Although this fit captures the average trend of the response, the estimated $pCO_{2}$ is always either over- or under-estimated due to the different rates for protonation/de-protonation, as mentioned above.

Given these data, we can propose a second model (Fit2 in Figure 5b), which also uses Equation (Equation 4), but with different calibration parameters (A,B,C,D) depending on whether the derivative of the fluorescence ratio is positive (CO_2_ increasing) or negative (CO_2_ decreasing). This model improves on the over-/under-estimation of Fit1 (higher $R^{2}=0.9808$) and follows closely the reference $pCO_{2}$ readings, as can be seen in Figure 5c and the residual plot in Appendix A. The standard deviation σ of the difference between the reference sensor and our $pCO_{2}$ estimate supports what is seen in Figure 5c, with $\sigma_{Fit1}=5.50$ mmHg and $\sigma_{Fit2}=2.73$ mmHg (see Appendix A for Bland–Altman plots of the models). Further, the results from Fit1 were found to lag behind the reference reading by 37 s, while Fit2 showed no lag.

The work in [25] proposed a linear dependence with CO_2_, with the fitting coefficients corresponding to the ratios of the different kinetic rates. A linear dependence does not capture the curvature of our data (see Appendix A), most likely due to the added complexity of CO_2_ diffusing through the different layers of the film.

The estimated sensitivity of our device considering the simpler linear relationship between *R* and $pCO_{2}$ yields a rate of change for R of:(5)$$\frac{\Delta R}{\Delta pCO_{2}}=\frac{\left(R^{max}-R^{min}\right)}{\left(pCO_{2}^{max}-pCO_{2}^{min}\right)}=0.13/\mathrm{mmHg}$$
or a percentage rate of change:(6)$$\frac{\Delta R\%}{\Delta pCO_{2}}=100\cdot \frac{\left(R^{max}/R^{min}-1\right)}{\left(pCO_{2}^{max}-pCO_{2}^{min}\right)}=14.2\%/\mathrm{mmHg}$$
obtained using the data from Figure 5.

In summary, our ratiometric readings and calibration algorithm are able to reliably obtain $pCO_{2}$ readings during multiple hours of measurements without being affected by photobleaching, and do so without deviating from the bench-top, commercial $pCO_{2}$ sensors. Depending on the application, longer monitoring times may be required, and the use of the films could be limited by HPTS photobleaching. This could be easily solved by periodically replacing films or reducing the sampling frequency of the device so the films are probed less frequently and, consequently, yield high signals for longer wear times.

## 4. Conclusions

In this work, we aimed to explore different HPTS-based ion pairs within different support matrices in order to identify optimal materials to be used with optical wearable devices for real-time monitoring of transcutaneous CO_2_ partial pressure. A thorough characterization of our materials, and most notably (HPTS)/(TOA)_4_ embedded within a PPMA matrix, showed them to be highly sensitive in the physiological CO_2_ range (0–50 mmHg). These films were found to be intrinsically insensitive to changes in humidity, a consideration that is important for skin-worn devices, as well as photostable during continuous sampling for long periods. We incorporated these materials into a medical-grade, multi-layer film to measure CO_2_ transcutaneously, carried out with an optical wearable device prototype. We proposed a detection and calibration methodology, relying on the formulation’s isosbestic point within the physiological CO_2_ range, which can robustly and reversibly detect CO_2_ changes within the physiological range, while compensating for fluorescence intensity changes due to photobleaching, as well as motion artifacts. The response speed of our films is comparable to commercial NDIR CO_2_ sensors and is ideal to detect physiological changes, which occur on a timescale of minutes. Our work shows great potential to develop commercial, miniaturized wearable devices to monitor tissue CO_2_ partial pressure at the skin surface. Further, both the device and the film structure, based on our team’s approach to measure transcutaneous oxygenation, have already been validated clinically, detecting changes in tissue oxygenation non-invasively in different physiological scenarios. Our prior work, along with the results shown here present an ideal platform to obtain dual O_2_/CO_2_ wearable sensors.

Future work will aim to develop a new scattering layer that is highly permeable to CO_2_ [32] in a wide temperature range so as not to require heating for transcutaneous applications. Alternatively, our devices could incorporate a small heating element to achieve temperatures around 42–45 °C, which are typically used in standard-of-care transcutaneous gas sensing devices. In the near future, we will aim to adapt our technology to create wearable or portable devices of different form factors, allowing us to continuously monitor both O_2_ and CO_2_ on the skin surface [26,33], in exhaled breath (capnography), and within arterial lines or muscle tissue [27].

## Figures and Tables

**Figure 1 biosensors-12-00333-f001:**
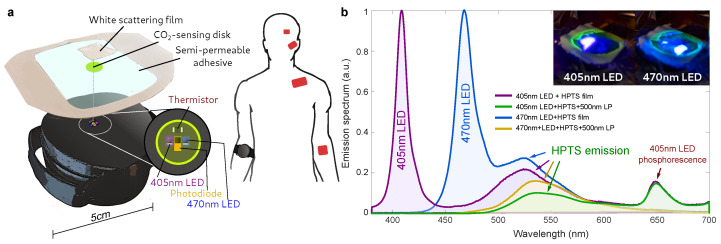
(**a**) Wearable device and $CO_{2}$-sensing film for continuous transcutaneous monitoring of $pCO_{2}$. The film emission is excited via two (405 nm and 470 nm) high-intensity LED’s and sampled via a 500 nm long-pass filter and a PIN photodiode. (**b**) Optical spectra of the two different excitation LEDs and the $CO_{2}$-sensing dye emission, as shown in the inset. The addition of a 500 nm long-pass filter removes the LED emission.

**Figure 2 biosensors-12-00333-f002:**
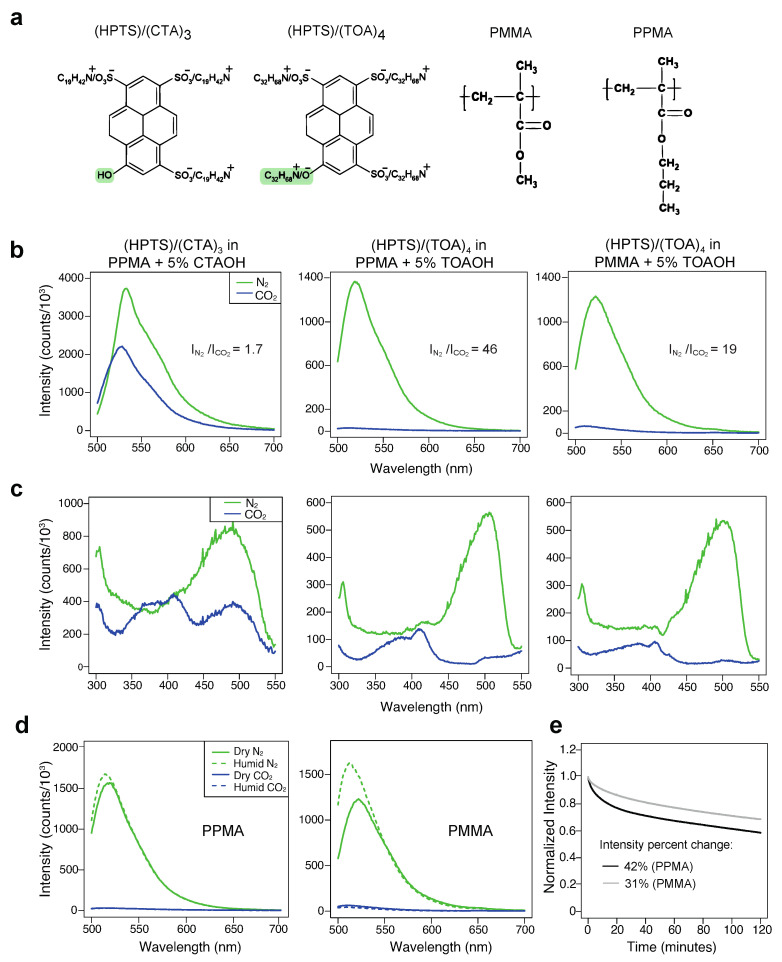
(**a**) Chemical structures of ion pairs and polymer matrices: (HPTS)/(CTA)_3_, (HPTS)/(TOA)_4_, poly(methyl methacrylate)(PMMA), and poly(propyl methacrylate)(PPMA). The pH sensitivity of the ion pairs arises from the highlighted functional groups. (**b**) Emission spectra of (HPTS)/(CTA)_3_ in PPMA and (HPTS)/(TOA)_4_ in PPMA and PMMA under CO_2_ and N_2_ conditions. (**c**) Excitation spectra (collected at 570nm) of (HPTS)/(CTA)_3_ in PPMA and (HPTS)/(TOA)_4_ in PPMA and PMMA under CO_2_ and N_2_ conditions. (**d**) Moisture sensitivity of (HPTS)/(TOA)_4_ in PPMA and PMMA under CO_2_ and N_2_ conditions. (**e**) Photostability comparison of (HPTS)/(TOA)_4_ in PPMA and PMMA under air condition.

**Figure 3 biosensors-12-00333-f003:**
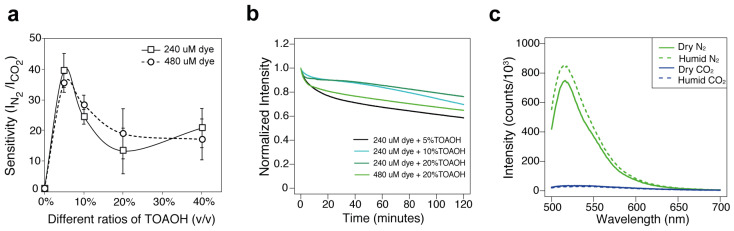
(**a**) Sensitivity of materials made of 240 μM or 480 μM (HPTS)/(TOA)_4_ in PPMA with the addition of 0%, 5%, 10%, 20%, and 40% (*v*/*v*) methanolic solution of TOAOH. (**b**) Photostability comparison of sensing films prepared from 240 μM (HPTS)/(TOA)_4_ in PPMA containing 5%, 10%, and 20% (*v*/*v*) TOAOH solution and 480 μM (HPTS)/(TOA)_4_ in PPMA with 20% (*v*/*v*) TOAOH solution under the air condition. (**c**) Moisture sensitivity of the material prepared from 240 μM (HPTS)/(TOA)_4_ in PPMA with 10% (*v*/*v*) TOAOH solution under CO_2_ and N_2_ conditions.

**Figure 4 biosensors-12-00333-f004:**
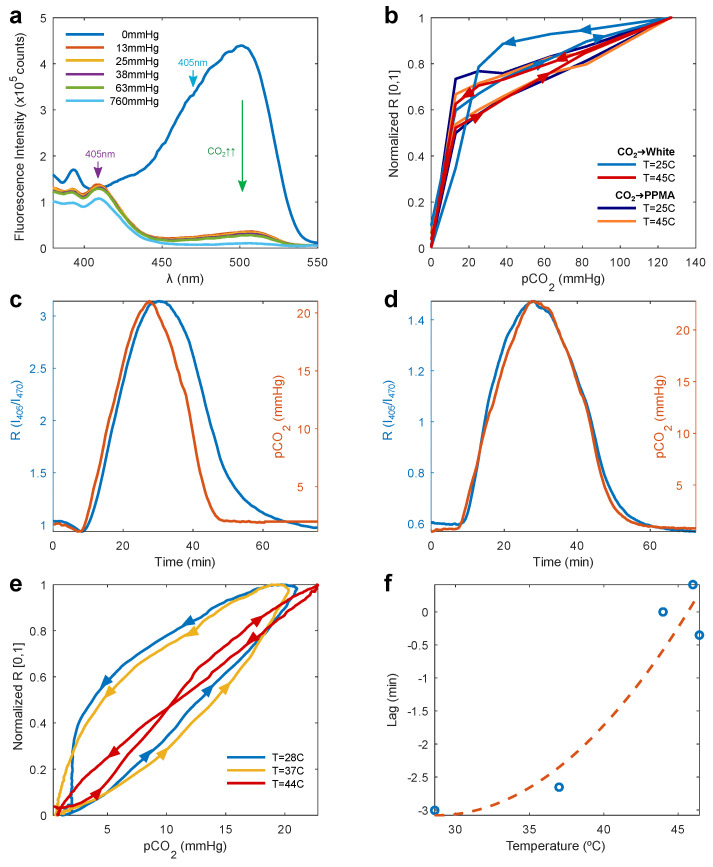
(**a**) Excitation spectra measured at 570 nm of the (HPTS)/(TOA)_4_ in the PPMA formulation exposed to different CO_2_ partial pressures. (**b**) Normalized *R* (between [0,1]) vs. CO_2_ partial pressure of a PPMA/white coating sample, showing a delayed diffusion of CO_2_ through the white coating (CO_2_ → white), which disappears at temperatures over 40 °C. The fluorescence ratio *R* measured with the wearable is highly sensitive to changes in CO_2_, with our prototypes showing a delayed response with respect to the reference $CO_{2}$ sensor at (**c**) T = 25 °C, attributed to CO_2_ diffusion through the white scattering layer, vanishing when heating up to (**d**) T = 44 °C. (**e**) Normalized *R* vs. CO_2_ for the wearable at different temperatures, with the delayed response vanishing at higher temperatures. (**f**) Time delay (lag) between our prototype’s signal and the reference CO_2_ sensor as a function of temperature.

**Figure 5 biosensors-12-00333-f005:**
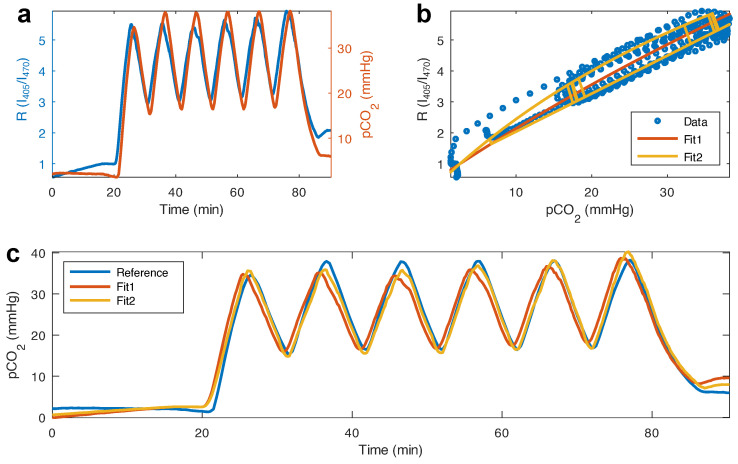
(**a**) Response of the film to changes in CO_2_, plotted along with a reference sensor’s CO_2_ readings. (**b**) Fit of two different calibration algorithms to the fluorescence ratio *R*, plotted as a function of the reference CO_2_. Fit1 considers a quadratic dependence on CO_2_, while Fit2 also considers a quadratic dependence on CO_2_, but with different coefficients depending on whether *R* (and hence, CO_2_) is increasing or decreasing. (**c**) Reference and estimated CO_2_ from our prototype, obtained with both algorithms.

## Data Availability

Study data can be made available upon request to the corresponding authors.

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
