# Peer review of "A Patient-Ready Wearable Transcutaneous CO2 Sensor"

_biosensors, 2022, doi:10.3390/bios12050333_

Round 1

Reviewer 1 Report

The authors reported the development of a wearable prototype device for transcutaneous CO2 monitoring based on quantifying the fluorescence of a highly breathable CO2-sensing film. The authors provided detailed methods and results on the materials development and calibration. The proposed wearable device has the potential to provide continuous monitoring of transcutaneous CO2 partial pressure.

The topic is significant; the paper provided detailed selection and optimization of sensing film compositions,  how TOAOH rations affected sensitivity, photostability, dark stability, and response and calibration of the prototype. Here are my comments, and I hope they can help improve the paper.

  1. The title stated a patient-ready wearable device, but there was no patient tested in this manuscript. It is not clear how the device will be worn from the paper. During the calibration, was the device worn by a patient? How the tightness and location of the device could impact the results?
  2. The abstract stated that the film's fluorescence is highly sensitive to changes in CO2 partial pressure. It would be better to provide the values of the sensitivity.
  3. How the movement of the wearable and moisture (for example, could be caused by sweating).

Reviewer 2 Report

The paper by Juan Pedro Cascales et al. reports a sensor that can be used to monitor CO2. It is beneficial to check the level of CO2 in a wearable and noninvasive platform. The data are informative. However, the manuscript should be revised to maximize the benefit to readers and cover significant aspects. Many errors in writing and English and typing styles. After revision, if the authors provide a detailed discussion and clarify the points and add improvement, it should be very interesting to consider for publication.

  1. Figure 1b: How did you know the peak at 525 nm was caused by HPTS emission? Why did the LED peak disappear when adding 500 nm LP?
  2. For Fig.2. It would be great to highlight the key functional group present in the selected molecules. The integration in the matrix should be also drawn.
  3. It would be clear to compare the spectra with/without the filter (considering the complexity using in the miniaturized system).
  4. For polymers, please add molecular weight for all.
  5. For the system to control N2/ CO2 gas mixture, please give more details about the equipment and geometry of the tool to attach the system to the sensor.
  6. “Optimization of sensing film compositions”
    • Figure 2: Why did you choose the ratio of I(N2) / I(CO2) as an indicator of the sensitivity of the CO2 sensor?
    • What was the mechanism for detecting CO2 in each sensing molecule composition and the support matrices?
  7. “The effect of TOAOH ratios on sensitivity, photostability, and dark stability”
    • Figure 3a: Why did various ratios of TOAOH (v/v) the concentration at 40% of the 240 M dye have a higher sensitivity than 480 M?
    • Why did the CO2 sensitivity decrease with the further increase of TOAOH 219 ratios?
  8. “Response and calibration of the wearable”
    • What was a gold standard reference? And how to prepare the gold standard reference?
    • Figure 4a: Why did the dye emission at 0 mmHg (the CO2 range of interest) different from the other pressures?
    • Figure 4e: Why was this graph at T=44 different from others?
  9. 5b. Please elaborate on the second-order polynomial model and different second-order polynomial. Compare with literature more, in addition to linear dependence. Please introduce the motivation to apply these two different modes. To be clear, please support the model by adding residual analysis too (i.e., residual plot). The equation with coefficient and R-square are needed.
  10. The product yield after the synthesis?
  11. Please comment on the potential crosstalk effect due to the fluctuation of O2 (as an interfering gas) for this sensor.
  12. For the hysteretic study, please mention the sweeping rate of varying temperatures.
  13. Any considerations on skin thickness/and skin color? Also, elaborate on your layer (in the wearable sensor).
  14. Please elaborate on biocompatibility and flexibility to comply with the human skin.
  15. For ‘Conclusions’, the challenges of this work that the authors have addressed should be emphasized. 
  16. The table comparing the merits of this work vs other existing arts should be interesting to emphasize the work.

Reviewer 3 Report

The manuscript entitled ‘A patient-ready wearable transcutaneous CO2 sensor’ describes a prototype device based on widely explored pH-sensitive fluorescent dye pyranine. The device's performance was evaluated in the controlled settings, and a robust detection and calibration methodology was proposed. The presented work undoubtedly would expand the implementation of CO2 sensors within wearable devices; however, several issues should be addressed to improve the quality of the manuscript.

Although HPTS was widely used in CO2 optical sensors, the principle of operation should be briefly described (e.g., scheme, equation) to make the data/ text more comprehensible. As such, the involvement of water molecules in the protonation/ deprotonation of the dye should be emphasized.

The organic base (phase transfer agent) TOAOH/ CTAOH was added to the polymer cocktail along with the ion pair to ensure the sensor’s response, as membranes without the base exhibited no response to CO2. The impact of lipophilicity of CTAOH and TOAOH on the sensor’s response was discussed. Here, the log P values for lipophilic cations could have been compared, 2.40 vs. 4.47. Also, the influence of different amounts of TOAOH (0-40%, v/v) on sensitivity was evaluated, but no hypothesis on the observed behavior was given. In previous works, an organic base was added to neutralize the acidic residues in the polymer and stabilize the deprotonated form of the indication ion pair within the matrix. Could that also be the case for the methacrylate matrix?

A dual excitation ratiometric method was used for the pCO2 sensing system. In the current form, the details of this approach are not precise and need further clarification. Typically, the relative emission intensity at a fixed wavelength (e.g., 515 nm) following excitation at λ1 and λ2 directly reflects the relative activities of the acid and base forms, and thus pH (or pCO2) (R= Iλ1/ Iλ2, where I stands for the fluorescence intensity measured e.g., at 515 nm). Here an integration was performed (p. 9, line 261) ‘I405 and I470 are calculated by integrating the spectra excited at 405 and 470 nm’.

In Fig. 3.a. the ‘sensitivity’ has been plotted against different ratios of TOAOH. Although the term ‘sensitivity’ has been defined within the text as the ratio of IN2/ICO2, it would be more clear for the reader to use the ratio.

The Fig. 4 caption is too extensive. It contains a discussion that should be moved to the main text. The axis titles can be confusing. Fig. 4.a. 0Y – excitation spectrum instead of fluorescence intensity, Fig. 4.c and d – Luminescence in [a.u.] appears. Is it a previously defined R-value? Also, the last sentence in Fig. 5 caption should be moved to the discussion section.

Round 2

Reviewer 1 Report

Thank you for the detailed responses and thorough edits. All my comments have been successfully addressed.

Reviewer 2 Report

It should be good for publication.